# A Bioactive Chitosan−Based Film Enriched with Benzyl Isothiocyanate/α−Cyclodextrin Inclusion Complex and Its Application for Beef Preservation

**DOI:** 10.3390/foods11172687

**Published:** 2022-09-02

**Authors:** Hongyan Wu, Xinying Ao, Jianan Liu, Junya Zhu, Jingran Bi, Hongman Hou, Hongshun Hao, Gongliang Zhang

**Affiliations:** 1School of Food Science and Technology, Dalian Polytechnic University, Dalian 116034, China; 2Liaoning Key Lab for Aquatic Processing Quality and Safety, Dalian 116034, China; 3Jinkui Food Science and Technology Corporation, Dalian 116033, China; 4Department of Inorganic Nonmetallic Materials Engineering, Dalian Polytechnic University, Dalian 116034, China

**Keywords:** benzyl isothiocyanate, α−cyclodextrin, chitosan, inclusion complex, composite film, antibacterial activity, food packaging

## Abstract

A bioactive packaging material based on chitosan (CS) incorporated with benzyl isothiocyanate (BITC) and α−cyclodextrin (α−CD) was fabricated to evaluate its preservative effects on fresh beef stored at 4 °C for 12 d according to the quality analysis. The Fourier-transform infrared (FTIR) spectrum revealed that the major structural moiety of BITC was embedded in the cavity of α−CD, except for the thiocyanate group. FTIR and X-ray diffraction analysis further verified that intermolecular interactions were formed between the BITC−α−CD and CS film matrix. The addition of BITC−α−CD decreased the UV light transmittance of pure CS film to lower than 63% but still had enough transparency for observing packaged items. The CS−based composite film displayed a sustainable antibacterial capacity and an enhanced antioxidant activity. Moreover, the total viable counts, total volatile base nitrogen, pH, thiobarbituric acid–reactive substances, and sensory evaluation of the raw beef treated with the CS−based composite film were 6.31 log colony-forming unit (CFU)/g, 19.60 mg/100 g, 6.84, 0.26 mg/kg, and 6.5 at 12 days, respectively, indicating the favorable protective efficacy on beef. These results suggested that the fabricated CS−based composite film has the application potential to be developed as a bioactive food packaging material, especially for beef preservation.

## 1. Introduction

The food−borne disease occurrence resulting from meat product deterioration has brought about enormous challenges to the meat industry [1]. On the other hand, inappropriate preservation−caused meat spoilage also led to food loss and waste issues [2]. In this context, the active food packaging technique, incorporating plant−derived preservatives into food package materials, has become a novel strategy to resolve these concerns and, meanwhile, to meet consumers’ demands for healthy and minimally processed foods [3].

Isothiocyanates (ITCs), one class of plant secondary metabolites generated from glucosinolates hydrolysis, are natural food preservatives in the Generally Recognized as Safe list [4] and possess versatile physiological properties including antibacterial, anti−inflammatory, and anticancer [5,6]. Among the ITCs, the aromatic sulfur flavor benzyl isothiocyanate (BITC) has grabbed plenty of attention because of its superior potential in suppressing a broad spectrum of food−borne bacteria [7,8]. However, the widespread application of this sulfur−containing compound as a native preservative in the food industry has been severely restricted because of its poor aqueous solubility, high volatility, and pungent odor [9,10].

In recent years, research on employing cyclodextrins (CDs) as the carriers to trap a variety of hydrophobic bioactive compounds forming inclusion complexes is being increasingly expanded [11,12]. The reasons underlying this trend lie in the unique structural characteristic of CDs, which possess a lipophilic central cavity and hydrophilic outer surface, facilitating to increase the inferior solubility of the included bioactive substance [13]. Alpha−cyclodextrin (α−CD) is one type of oligosaccharide consisting of six glucose units [14], the inclusion property of which has been proved application potential in food, medicine, and cosmetics [15]. α−CD encapsulation of moringa isothiocyanate promoted aqueous solubility and stability, which in turn exhibited a higher efficacy in suppressing the neuroblastoma cell proliferation than solely applying the ITC compound [16]. Furthermore, the moringa isothiocyanate−α−CD complex with enhanced solubility exhibited favorable antibacterial efficacy against *Staphylococcus aureus* sensible strains as well as the bactericidal effect against clindamycin−resistant strains [17]. Moreover, the ally isothiocyanate (AITC)−α−CD complex obscured the inherent pungent odor and ensured its sustainable release in the medium [18]. Therefore, α−CD could be expected as a promising carrier to overcome the shortcomings of ITCs. Up to date, there are no studies regarding fabricating inclusion complex by trapping BITC into the α−CD cavity to the best of our knowledge.

Conventional petroleum−based plastic materials have been utilized in food packaging for a couple of decades, whereas the resulting environmental pollution and the growing demands for green foods promote the research on developing natural biopolymer−based films [19]. Encouraged by this premise, natural raw materials such as chitosan (CS), a cationic polysaccharide derived from partly deacetylation of chitin, has been considered an applicable candidate as the edible film [20]. The favorable biological features of CS include non−toxic, bio−degradable, bio−compatible, excellent film−forming capacity, and a certain degree of antibacterial potential [21]. However, pure CS film is deficient in the good barrier capacity, potent antibacterial activity, and antioxidant ability, which restricts its practical applications in the food packaging industry [22]. The addition of *Carica papaya* seed essential oil, in which the BITC was the predominant constitute, into the CS/polyvinyl alcohol blend film has been corroborated to not only exhibiting excellent UV−blocking capacity but also favorable antibacterial efficacy [23]. Despite the CS/κ−carrageenan composite film incorporating AITC still presenting a mild pungent smell, it was useful to repress the growth of *Campylobacter jejuni* inoculated on the chicken meats and prolong the shelf−life [24]. CS−AITC coating in combination with organic acid application notably suppressed the colony growth rate of *Salmonella enterica* inoculated in grape tomato [25]. Furthermore, previous studies also suggested that adding bioactive compound and CD composed inclusion complex to the CS film−forming solution is beneficial to endow the pure film with enhanced functional and physical attributes [26,27].

Herein, α−CD was used to encapsulate BITC and the formed inclusion complex was further incorporated into the CS−based film. The structure characterizations were investigated using Fourier−transform infrared (FTIR) combined with X-ray diffraction (XRD) technologies. Moreover, the microstructure of the fabricated composite film was measured with aid of scanning electron microscopy (SEM). The effects of BITC−α−CD on the light transmittance, water barrier property, antioxidant capacity, as well as antimicrobial activity of the pure CS film were examined. Furthermore, beef was utilized as the fresh meat model to assess the freshness−retaining efficacy of the fabricated composite film.

## 2. Materials and Methods

### 2.1. Materials

In this case, α−CD (98%) as well as the aromatic sulfur−containing compound BITC (purity higher than 99%) were obtained from Sigma−Aldrich (St. Louis, MO, USA). CS (degree of acetylation higher than 95%) was obtained from Macklin Biochemical Co., Ltd. (Shanghai, China). Dimethyl sulfoxide (DMSO), acetic acid, and glycerol were procured from Tianjin Damao Chemical Co., Ltd. (Tianjin, China). *Salmonella enterica* serovar *Typhimurium* (*S*. *Typhimurium*, ATCC 14028) was obtained from Culture Collection (Beijing, China). The beef was obtained from the local supermarket (Dalian, China). Other reagents used in the current study were all of the analytical grade.

### 2.2. Fabrication of Inclusion Complex

In our preliminary experiments, the highest encapsulation efficiency ratio was obtained when the ratio of BITC to α−CD was 2 to 1 and the magnetic stirring temperature was 65 °C. Therefore, to obtain the inclusion complex at a 2:1 (guest: host) molar ratio, 6.8 M of BITC anhydrous ethanol solution (0.6 mL) was gently added to the 20.6 mM of α−CD solution at 65 °C under magnetic stirring for 3 h. The obtained solution was ultrasonically homogenized for 0.5 h, followed by freezing at −80 °C and vacuum freeze−drying (Figure 1).

### 2.3. Fabrication of the CS−Based Composite Film

The CS−BITC−α−CD composite film was fabricated utilizing the solution casting method. Briefly, 2% (*w/v*) CS was added to 1% (*v/v*) of the acetic acid solution, followed by magnetic stirring at 60 °C for 1 h. After completely dissolving, 1% (*w/v*) of BITC−α−CD was added to the CS solution and stirred for 0.5 h, and then 2.5% (*v/v*) of glycerol was added to continue stirring for 0.5 h. Subsequently, the mixture was ultrasonically treated for 2 h to remove bubbles, followed by gently pouring into the mold for casting. The poured solution was dried at 40 °C for 6 h and then kept at ambient temperature for 2 days (Figure 1). The pure CS film without the addition of BITC−α−CD was used as the control.

### 2.4. Characterization of Inclusion Complex

#### 2.4.1. FTIR Analysis

The sample of α−CD or BITC−α−CD (1 mg) was ground with 100 mg of KBr to prepare the tablet and was determined by the FTIR spectrum (Norwalk, CT, USA) at the range of 4000–650 cm^−1^ with 4 cm^−1^ of resolution.

#### 2.4.2. XRD Analysis

The physical states of α−CD and BITC−α−CD were examined by the X-ray diffractometer (Shimadzu, Kyoto, Japan). The XRD machine was equipped with a Cu−Κα radiation source operating at a voltage of 40 kV and a current of 40 mA. The 2θ scan range was set from 10° to 60°. The crystallinity index (CI) of the test film sample was calculated with the following Equation:(1)$$\mathrm{CI}\;(\%)=\frac{\mathrm{Area}\;\mathrm{of}\;\mathrm{crystalline}\;\mathrm{peaks}}{\mathrm{Area}\;\mathrm{of}\;\mathrm{crystalline}\;\mathrm{peaks}\;+\;\mathrm{Area}\;\mathrm{of}\;\mathrm{amorphous}\;\mathrm{peaks}}\;\times \;100$$

### 2.5. Characterization of the CS−Based Composite Film

The FTIR spectra and XRD patterns of pure CS and CS−BITC−α−CD composite films were determined as same as the analysis of BITC−α−CD.

### 2.6. Appearance, Light Transmittance and Surface Morphology of the CS−Based Composite Film

#### 2.6.1. Appearance

The appearance images of pure CS and CS−BITC−α−CD composite films were captured under the same background and light conditions.

#### 2.6.2. Light Transmittance

The barrier capacity of the test film samples against UV and visible light was determined using a UV−vis spectrophotometer (Cambridge, MA, USA) with a scanning range of 200–800 nm.

#### 2.6.3. Surface Morphology

The surface of the pure CS and CS−BITC−α−CD composite films was observed by the scanning electron microscopy (SEM) (Hitachi, Tokyo, Japan) operating at an accelerating voltage of 5 kV and a magnification of × 200. All film samples were coated with gold for 10 s before observation.

### 2.7. Moisture Content (MC), Water Solubility (WS), and Water Vapor Permeability (WVP) of the CS−Based Composite Film

The tailored film samples (1 cm × 2 cm) were initially weighed as W_0_ and then dried at 105 °C to obtain the constant weight W_1_. Subsequently, the dried film samples were immersed in 50 mL of distilled water for 24 h and dried at 105 °C until obtaining the constant weight (W_2_) [28]. The MC and WS of the test film samples were calculated according to the following Equations:(2)$$\mathrm{MC}\;(\%)=\frac{W_{0}-W_{1}}{W_{0}}\;\times \;100$$
(3)$$\mathrm{WS}\;(\%)=\frac{W_{1}-W_{2}}{W_{1}}\times \;100$$

The WVP of the test film samples was examined followed the protocol reported previously [12] with a few modifications. The test films were fixed at the top of the specific vessel, which contained 3 g of anhydrous CaCl_2_ to provide 0% of room humidity (RH). Subsequently, the vessel covered with the test film sample was placed in a desiccator with 75% RH. The vessel was weighed at an interval of 24 h for three days. The WVP was thereby calculated according to the following Equation:(4)$$\mathrm{WVP}\;(g/m\cdot s\cdot \mathrm{Pa})=\frac{\Delta m\times L}{A\times \Delta t\times \Delta P}$$
where Δm is the weight variation (g) of the test vessel, L is the thickness (m) of the sample, A is the permeation area (m^2^), Δt is the storage time (s), and ΔP is the vapor water pressure difference of the two sides of the test film.

### 2.8. Antibacterial Activity of the CS−Based Film

The antibacterial capacity of CS−based films was carried out based on the previously reported protocol with minor modifications [29]. In brief, *S*. *Typhimurium* was cultured in the tryptic soy broth (TSB) for 12 h to reach the logarithmic growth period (bacterial concentration around 10^9^ CFU/mL). The diluted cultivated *S*. *Typhimurium* suspensions and 0.9 g of film pieces were added to the broth media, incubating for 24 h. The bacterial growth was determined at OD_600_ nm at an interval of 12 h. The antibacterial efficiency was expressed as the percentage over the corresponding control.

### 2.9. Antioxidant Activity of the CS−Based Film

The free radical scavenging capacity of the CS−based films was determined based on the protocol reported previously [30] with a few modifications. Briefly, 0.1 g of test film samples was added to 10 mL of ethanol and magnetic stirring at ambient temperature for 24 h to acquire the film extract solution. Subsequently, 3 mL of the film extract solution was quickly mixed with 3 mL of 0.1 mM DPPH ethanol solution. The obtained mixture was placed in the dark for 0.5 h. The examination of the mixture absorbance was carried out at 517 nm by the microplate reader (Madison, WI, USA) and the DPPH scavenging capacity was calculated using the following formula:(5)$$\mathrm{DPPH}\;\mathrm{scavenging}\;\mathrm{activity}\;(\%)=\frac{A_{\mathrm{blank}}-A_{\mathrm{sample}}}{A_{\mathrm{blank}}}\;\times \;100$$
where A_blank_ and A_sample_ means the absorbance of the reaction solution without and with the test film sample, respectively.

### 2.10. Preservative Effects of the CS−Based Composite Film on Fresh Beef

#### 2.10.1. Preparation of Beef Treated by the CS−Based Composite Film

The beef samples were cut into cubes (5 g), followed by wrapping with the same size of polyethylene terephthalate (PET) film, CS film, or CS−BITC−α−CD composite film. Due to the short shelf−life of fresh meat products (generally three to five days) [31], twelve days were defined as the longest storage time. All samples were stored at 4 °C and taken out for freshness index analysis at the indicated time points (0, 4, 8, and 12 days). The beef wrapped with commercially used PET film was used as the control.

#### 2.10.2. Beef Quality Analysis

##### Total Viable Counts (TVC) Determination

Five grams of beef samples at different storage time points were mixed with 45 mL of sterilized normal saline (0.85%) in aseptic bags and homogenized at 10,000 rpm for 1 min using a Stomacher blender (HBM Biomed, Tianjin, China). The mixture was then tenfold serial diluted using the sterile saline solution. A 0.1 milliliter of appropriate dilution was plated on the plate count agar at 37 °C for two days for counting [30].

##### Total Volatile Base Nitrogen (TVB−N) Determination

Five grams of beef samples at different storage time points were homogenized in 25 mL of distilled water for 30 min. The mixture was filtered, and 5 mL of supernatant was then mixed with 5 mL of magnesium oxide solution (10 g/L). Subsequently, the mixture was distilled for 5 min with the assistance of an automatic Kjeldahl distillation unit. The distillate was collected by 5 mL of boric acid solution (20 g/L) with 6 drops of mixed indicators. Finally, the boric acid solution was titrated with the 0.01 mol/L of hydrochloric acid solution and the results were expressed as mg/100 g beef sample.

##### pH Determination

Five grams of beef samples at different storage time points were ground and mixed with 45 mL of deionized water, followed by homogenization for two minutes. The homogenate was placed at room temperature for 0.5 h and then determined by a pH meter (Matthaus GmbH, Poettmes, Germany) [32].

##### Thiobarbituric Acid−Reactive Substances (TBARS) Determination

Five grams of beef samples at different storage time points were mixed with 20 mL of 7.5% (*w/v*) trichloroacetic acid solutions and centrifuged at 6000 rpm for 1 min, followed by filtering. The obtained filtrates were incubated with 5 mL of 0.02 M (*w/v*) 2−thiobarbituric acid (TBA) at 95 °C for 2 h. The final reaction solutions were cooled down and then measured the absorbance at 532 nm [33]. The calibration curve was established using 1,1,3,3−tetraethoxypropane at the concentrations ranging from 0.0125 mg/kg to 1.25 mg/kg and the results were expressed as mg/kg of beef sample.

##### Sensory Evaluation

The sensory evaluation (color, viscosity, odor, visible impurity, and texture) of beef samples was carried out following the previous report with minor modifications [34]. Ten trained assessors from Dalian Polytechnic University (Dalian, China) executed the project based on a 10−point scale (10 = best quality, 5 = acceptable limit, 1= worst quality).

## 3. Results and Discussion

### 3.1. Characterization of CS−Based Composite Film

FTIR analysis is a common tool to investigate the interactive behavior among the components in the fabricated film [35]. As shown in Figure 2a, in accordance with recent studies [36,37], α−CD showed characteristic peaks at 3400 cm^−1^ (−OH symmetric stretching vibration), 2926 cm^−1^ (C−H stretching vibration), 1156 cm^−1^ (C−O−C glycosidic bridge stretching vibration), and 1028 cm^−1^ (C−O stretching vibration). BITC−α−CD displayed a quite similar spectrum to that of α−CD, except for the existence of characteristic peaks at 2140–2040 cm^−1^. In recent studies, BITC was reported to have absorption bands at 2140–2040 cm^−1^ attributed to the thiocyanate group and at 1600–1345 cm^−1^ ascribed to the benzene ring [10,38]. Moreover, the benzene ring of BITC was suggested to be entrapped in the cavity of γ−CD, whereas the thiocyanate moiety was exposed outside [38]. Another study entrapping BITC to β−CD reported a resemble inclusion complex pattern [39]. Collectively, these results indicated that although the thiocyanate group of BITC is exposed out of the cavities of CDs, the major structural moiety of BITC could be completely embedded in the channels of CD compounds. For pure chitosan, the feature absorption bands appeared at 3293 cm^−1^ (stretching vibrations of the N−H group and −OH group), 2922 cm^−1^ (asymmetric as well as the symmetric vibrations of the C−H group), 1647 cm^−1^ (amide I), 1554 cm^−1^ (amide II), and 1028 cm^−1^ (C−O stretching vibration) [11,28,40,41]. After adding BITC−α−CD to the CS film matrix, slight wavelength shifts of these feature bands were observed. For instance, the amide II shifted from 1554 cm^−1^ to 1562 cm^−1^, indicating that some hydrogen bonds in the CS film matrix were disrupted because of the complete incorporation of BITC−α−CD. Moreover, the stretching vibrations of the N−H as well as −OH stretching bands at 3293 cm^−1^, amide I at 1647 cm^−1^ and C−O at 1028 cm^−1^ shifted to 3303 cm^−1^, 1651 cm^−1^ and 1034 cm^−1^, respectively, and slightly increased their intensities with the addition of BITC−α−CD. These results suggested that certain intermolecular forces, such as hydrogen bond crosslinks or non−covalent interactions [28], were formed between CS and BITC−α−CD.

XRD analysis is a common and desirable method to measure and characterize the crystal structures of the test samples [42]. Since BITC under ambient temperature is at aqueous status, the XRD information of which is difficult to acquire [38]. The diffraction patterns of α−CD and BITC−α−CD were displayed in Figure 2b. The sharp and intense peaks of α−CD at 2θ of 11.96°, 14.26°, 15.84°, 19.2°, and 21.68° were in line with the previous report [43], which validated the crystalline nature of this substance. However, BITC−α−CD exhibited a different diffraction pattern, with broad and intense peaks at 2θ of 11.26°, 12.88°, 16.18°, 19.84°, and 22.62°. Furthermore, the relative crystallinity reduced from 66.28% to 51.80% with the addition of BITC. These findings suggested that the spectra of BITC−α−CD are not a simple superposition of the pure spectra of BITC and α−CD, indicating the formation of an inclusion complex. For pure CS film, it exhibited the primary peak at 2θ = 20.48° and the relative crystallinity was 11.10%, reflecting its amorphous status [28]. Moreover, the crystallinity of BITC−α−CD was reduced after being incorporated to the CS film with the relative crystallinity of 19.74%, indicating the formation of amorphous complex between CS and BITC−α−CD. These findings were in accord with the FTIR results (Figure 2a), suggesting that BITC−α−CD could be completely incorporated to the CS film matrix by physical interactions. In addition, it was reported that dense and amorphous characteristics rendered alumina−based films with excellent barrier capacity against high temperature [44]. The addition of amorphous poly (lactic acid) (PLA) to the poly (butylene succinate−*co*−butylene adipate) film matrix endowed the composite film with enhanced gas barrier property [45]. Therefore, we speculated that the amorphous status of CS−based film might be beneficial to the barrier capacity against certain environmental factors, such as temperature and oxygen.

### 3.2. Appearance, Light Transmittance and Surface Morphology of the CS−Based Composite Film

The appearance of the film is a crucial factor that determines whether consumers would accept the packed foods. As shown in Figure 3a, the pure CS film was yellowish and transparent. No significant difference in the color and transparency was seen from the composite film (Figure 3b), which preliminarily indicates that the incorporation of BITC−α−CD would not bring about alterations to the appearance of the CS film matrix. To better illustrate the physiochemical properties of the pure CS and CS−BITC−α−CD composite films, the color measurements, particle size measurements, surficial roughness, and the distribution of BITC−α−CD in the CS film matrix will be performed in our future studies.

UV−vis light irradiation may induce oxidative process of food, which results in food deterioration and thereby adversely influence organoleptic as well as nutritional qualities of food [46]. Therefore, favorable UV−vis light barrier capacity is of great importance for the foodstuff packaging material. As shown in Figure 3c, both pure CS film and composite film displayed an absorption peak below 250 nm, which might be related to the chromophores that exist in CS [32]. The transmittance of pure CS film was 21.42–80.43% in the UV spectra range (200–400 nm). After incorporating BITC−α−CD to the CS film matrix, the transmittance values at the same wavelength range were significantly decreased to lower than 63%. Moreover, the composite film still exhibited a relatively lower transmittance range (40.98–96.93%) in the visible spectra range (400–800 nm) compared to that of the pure CS film (75.42–99.14%). These observations indicated that addition of BITC−α−CD endowed CS film with improved UV−vis light barrier capacity at the expense of slightly reduced transparency. The previous study encompassing BITC to the CS/cellulose nanofibers composite film demonstrated that despite the transparency of the fabricated composite film is compromised, it was still capable to clearly observe the packaged items [46]. Therefore, combining with the appearance observations (Figure 3b), we concluded that CS−BITC−α−CD film might be beneficial to avoiding UV−vis light−caused oxidative deterioration and still possess enough transparency for consumers to observe foodstuffs.

To further examine the morphological change brought by BITC−α−CD to the CS film matrix, the surface morphology of the fabricated film was determined by SEM. As shown in Figure 3d, the pure CS film displayed a uniform and smooth surface without any porous and/or cracked structure. After adding BITC−α−CD, several light bumps evenly appeared in the composite film (Figure 3e). These observations suggested that the BITC−α−CD dispersed in the composite film solution in an even manner, indicating the favorable compatibility between BITC−α−CD and CS film−forming solution. Moreover, no large agglomerations in the microscopic surface of CS−BITC−α−CD composite film suggested that BITC−α−CD can be well dispersed in the CS film matrix due to intermolecular forces, which might be beneficial to improve the film strength [47]. The previous study reported that the uniform distribution of cellulose nanocrystals in the CS film effectively enhanced the physical and mechanical properties of the CS film matrix [48].

### 3.3. MC, WS, and WVP of the CS−Based Composite Film

In the present study, we investigated the effects of BITC−α−CD at different concentrations (100 μM, 150 μM, and 200 μM) on the moisture content, water solubility, water vaper permeability, antibacterial activity and antioxidant activity of the CS film. However, insignificant changes in these indices were observed. Therefore, the lowest concentration 100 μM was selected in the present study. MC is an important characteristic parameter reflecting the occupation of water molecules on the microstructure of the fabricated film [28]. As shown in Table 1, the MC of the CS−based composite film was slightly lower than that of pure CS film. Combining with the FTIR and XRD analysis (Figure 2), this might be ascribed to the fact that the interactions formed between BITC−α−CD and CS molecules occupy the hydroxyl and amino groups in the CS film matrix and thereby prevent their binding with water molecules. Similarly, Bai et al. [49] also found that the interactions between epichlorohydrin−β−cyclodextrin inclusion complex and CS film matrix caused a slight decrease of MC in the fabricated composite film. Furthermore, in comparison with the WS level of pure CS film (41.4%), the value slightly increased to 48.3% in the CS−BITC−α−CD composite film, which might be associated with the outer hydrophilic nature of α−CD [50]. Narasagoudr et al. [51] also found the WS level in the CS/poly (vinyl alcohol) matrix increased to 81.02% with the addition of the hydrophilic component and attributed this phenomenon to the moisture adsorption capacity of the incorporated component. Moreover, the WS of the CS film increased to 73.9% with the incorporation of hexahydro−β−acid/2−O−methyl-β-cyclodextrin inclusion complex [52]. Taken together, the comparatively lower WS value in the CS−BITC−α−CD film might indicate its better stability in the aqueous surroundings.

WVP is another important indicator to evaluate the moisture exchange rates between the internal packaged item and the external environment [53]. With the incorporation of BITC−α−CD, the WVP slightly decreased from 4.2 ± 0.1 g/m·s·Pa in the pure CS film to 3.9 ± 0.1 g/m·s·Pa in the composite film. In fact, the change of WVP index of the film depended on many factors, for instance, microstructures, density of the matrices, hydrophilicity, crystallinity, as well as interactions between polymeric components [54,55]. FTIR and XRD spectra indicated the interactions between CS and BITC−α−CD and an enhancing effect of BITC−α−CD on the relative crystallinity of CS matrix (Figure 2). SEM images indicated the modification of BITC−α−CD on the CS matrix microstructure (Figure 3d,e). No significant changes of MC and WS values (Table 1) indicated that the hydrophilic property of CS film matrix might be related to WVP insignificant variations. Moreover, the density of CS film is another important reason that causes the insignificant changes in WVP index. Wang et al. [56] revealed an insignificant decrease of WVP in the CS film enriched with *Lycium barbarum* fruit extract. They ascribed the insignificant variation to the higher density of the CS film matrix, which provided less interstitial space for water molecules to pass through. On the other hand, it was worth noting that the WVP value of CS−BITC−α−CD was lower than the previous reported CS/chondroitin sulfate/proanthocyanidin film (6.4 × 10^−11^ g/m·s·Pa) [32], CS/cellulose nanofiber/benzyl isothiocyanate film (1.2 × 10^−11^ g/m·s·Pa) [46], and CS/gelatin/quercetin film (7.6 × 10^−8^ g/m·s·Pa) [57], suggesting that the fabricated CS−BITC−α−CD had the desirable water vapor barrier ability and developing potential as the foodstuff packaging material.

### 3.4. Antibacterial and Antioxidant Properties of the CS−Based Composite Film

Incorporating the bioactive compound with antibacterial property into the packaging materials could help to ensure food safety by slowly releasing the antibacterial substance [58]. *S*. *Typhimurium* has been globally reported as the predominant foodborne pathogen contaminating eggs and meat products, especially beef [59]. Therefore, we used *S*. *Typhimurium* as the test model strain and examined the antibacterial potential of CS−based composite film against it. As shown in Table 2, although 12−h co−incubation with pure CS film displayed a slight growth inhibition against *S*. *Typhimurium* (4.2%), the inhibitory efficacy was decreased after 24 h co−incubation (1.3%). The antibacterial potential of CS could be owed to the interaction formed between its positively charged amino groups and the negatively charged macromolecules on the bacterial surface, which interferes with the organism metabolism process and even results in the leakage of intracellular content [60]. Contrarily, the CS−based composite film exhibited a time−dependently inhibitory capacity on bacterial growth, with a growth inhibition rate rising from 31.0% at 12 h to 53.3% at 24 h. The stronger antibacterial efficacy exhibited by CS−based composite film might be due to the excellent antibacterial potential of the added BITC. Previous studies indicated that BITC could prevent the growth of *S*. *Typhimurium* by disrupting the cell integrity, interfering with the bacterial energy metabolism process, and decreasing the mRNA levels of virulence−associated factors [7,8]. In addition, the sustainable antibacterial capacity of CS−based composite film also suggested that encapsulation of BITC in α−CD and then being incorporated into the CS film matrix might contribute to the controlled release of BITC. Resemble controlled−release efficacy of antibacterial compounds has also been reported in other CS−based materials [61,62].

The packaging material possessing antioxidant potential would contribute to reducing and preventing the occurrence of lipid oxidation in foods [62]. DPPH assay is a common tool to examine the direct antioxidant ability of the test sample. The mechanism of this assay is based on the fact that DPPH radical could adopt the hydrogen atom or the electron from the antioxidant agent [63]. The antioxidant capacities of pure CS and CS−based composite films were shown in Table 2. The DPPH free radical scavenging capacity of pure CS film was 11.4%, which was similar to previously reported 5% [64] and 12.5% [65]. The antioxidant ability of pure CS might be due to its free amino groups (−NH_2_), which are ready to react with DPPH radicals [66]. The addition of BITC−α−CD significantly enhanced the free radical scavenging capacity of the pure CS film. The improved antioxidant capacity might be related to the methylene group (−CH_2_) adjacent to the sulfur atom in the BITC, which is considered to react with free radicals by hydrogen atom transfer [67].

### 3.5. Preservative Efficacy of the CS−Based Composite Film on the Beef

The initial TVC value of the raw beef was around 3.34 ± 0.41 log CFU/g, which was comparable to the data proposed by Huang et al. [68], indicating the fresh quality of the test beef sample. A gradually upward trend in colonies was observed in all samples, with the increase rate following the order of PET film > CS film > CS−based composite film (Figure 4a). The allowable detection limit in fresh beef advocated by the International Commission on Microbiological Specifications for Foods is 7 log CFU/g [69]. It could be observed that the TVC values in PET film (6.72 ± 0.20 log CFU/g) and CS film (6.67 ± 0.35 log CFU/g) wrapped samples on the 8th day of storage were quite close to the detection limit and both exceeded it on the 12th day. For the CS−based composite film, the TVC value on the 12th day (6.31 ± 0.30 log CFU/g) was still lower than the limit. These results indicated that the shelf−life of beef wrapped with PET or CS film is no longer than 12 days, whereas the incorporation of BITC−α−CD into the CS film matrix effectively prolongs the refrigerated storage of raw beef, possibly related to the slow release of the antibacterial agent BITC. Similarly, Katekhong et al. [70] also found the inhibition of microorganisms in meats packaged with films containing nitrite, which extended the shelf−life of pork by more than 12 days. Qin et al. [71] revealed that the CS film enriched with tea polyphenol displayed microbiological shelf−life extension of meats up to 12 days.

The value of TVB−N, arising from protein as well as amine decomposition, is deemed to be a cardinal indicator reflecting the freshness of meats [72]. The TVB−N (mg/100 g) change of beef preserved at 4 °C for 12 days was shown in Figure 4b. After storing for 4 d, the TVB−N values of beef wrapped with PET film (16.80 ± 0.30 mg/100 g) and pure CS film (16.60 ± 0.40 mg/100 g) both surpassed the fresh meat standard (15 mg/100 g) [73]. However, the TVB-N in the CS−based composite film group exceeded the standard on the 8th day and gradually increased to 19.60 ± 1.50 mg/100 g on the 12th day. Compared with the PET film group, the slightly slower TVB−N growing tendency in the pure CS film might be ascribed to its antimicrobial potential, which retards the microorganism−caused protein and nitrogen−containing substance decomposition [74]. Moreover, the reduction of TVB−N level in beef sample wrapped with CS−based composite film can be related to the favorable antibacterial potential of BITC−α−CD, as we demonstrated in Table 2. Recently, Afshar Mehrabi et al. [75] also reported a retard increase of TVB−N level for CS film enriched with *Nepeta pogonosperma* extract packaged chicken meats during 4 °C storage and interpreted this phenomenon as the antibacterial property of the bioactive extract. Furthermore, the increment of the TVB−N value was also reported to be in good connection with the accelerated oxidation in beef samples [76]. Ruan et al. [77] investigated the effect of edible biopolymeric film containing epigallocatechin gallate and found that the delaying oxidation effect of the composite film was beneficial to the reduction of TVB−N in pork. Thus, combined with the results illustrated in Table 2, we deduced that the most favorable efficacy for controlling TVB−N displayed by CS−based composite film might be due to the combination of improved antibacterial and antioxidant potential.

As shown in Figure 4c, with the storage time prolonged, the pH values of all groups increased. This might be attributed to the accumulation of alkaline substances, which are generated by the microorganisms and endogenous enzymes−induced protein degradation [77]. Generally, the pH value of a meat sample higher than 6.7 is considered spoiled [78]. The pH increase rate of the beef sample wrapped with PET film was the fastest, surpassing the limit within 4 d. The pure CS film group also exceeded 6.7 on the 4th day. On the contrary, the pH in the CS−based composite film group was still lower than the limit on the 8th day and reached 6.84 ± 0.08 on the 12th day. Consistent with our results, Liu et al. [79] demonstrated that the pH of chilled meats treated with CS−based packaging film enriching with antibacterial 3−phenyllactic acid changed slowly during 4 °C storage. Alirezalu et al. [80] found that in comparison with the sole application of PET film or CS film, the CS−based film impregnated with ε−polylysine more effectively retarded the increment of pH values in beef, possibly associated with the prevention of microbial growth−induced alkaline substance accumulation. It can be therefore concluded that the antibacterial potential of the bioactive compound added might partly contribute to maintaining the pH value changes.

The TBARS value, changing from the secondary products of lipid hydroperoxides and peroxides, was applied to assess the degree of lipid oxidation in meats [33]. As shown in Figure 4d, the initial TBARS values in all groups were around 0.03 mg/kg and subsequently rose with the storage time prolonged because of the lipid oxidation caused by oxygen attack. The TBARS of beef samples wrapped with CS−based composite film on the 12th day was 0.26 ± 0.01 mg/kg, which was lower than PET and pure CS film groups. This result was in accord with the results of the free radical−scavenging capability assay (Table 2), which further verified the antioxidant potential of the composite film in practical applications. Du et al. [81] also demonstrated that CS−based film incorporating with rosemary extract displayed inhibitory effects on the lipid oxidation in grass carp samples. Hu et al. [82] revealed a reduction in the TBARS value changes in pork meats wrapped with the CS/starch aldehyde−catechin composite film.

The scoring results of beef samples treated with different types of films were presented in Figure 4e. The meat sample with a sensory score below 5 is considered unacceptable to consumers. During the storage period, the sensory scores of all beef samples decreased as the storage time increased. The deterioration rate of the meat sample wrapped by PET film was the fastest, with the sensory score significantly decreased to 3.5 from the 8th day of refrigeration. However, the sensory scores of beef samples wrapped with CS film or CS−based composite films on the 8th day of storage were 5.5 and 7, respectively, which were still in the acceptable range. With the prolonged storage time, the sensory score of the sample wrapped with CS film decreased to 4.5 on the 12th day, whereas the score in the composite film−wrapped group (6.5) was still acceptable until the last day of storage. Resemble results have been reported by applying CS−starch film enriching with pomegranate peel extract wrapping beef, which remained the sensory score in the acceptable range during 16 d of storage [83]. Moreover, the superior sensory performance of CS−based composite film was in good accordance with the results obtained from the TVC, TVB−N, pH, and TBARS determinations, reflecting the eminent preservation potential of CS−based composite film in raw beef.

## 4. Conclusions

This study encapsulated BITC using α−CD and further incorporated the inclusion complex into the CS film matrix to prepare a novel bioactive packaging film for raw beef preservation. The major structural moiety of BITC was completely embedded in the α−CD cavity, except for the thiocyanate group. BITC−α−CD was homogeneously incorporated into the CS film matrix through physical interactions. The addition of BITC−α−CD significantly enhanced the antibacterial efficacy, antioxidant potential, and UV−vis light barrier capacity of the pure CS film. Moreover, the CS−based composite film effectively prolonged the storage life of raw beef samples by slowing the increase of TVC, TVB−N, pH, and TBARS indexes, and meanwhile remaining the sensory characteristics. Collectively, the current study provides evidence demonstrating that the fabricated CS−based composite film has the application potential to be developed as a bioactive food packaging material, especially used for beef preservation.

## Figures and Tables

**Figure 1 foods-11-02687-f001:**
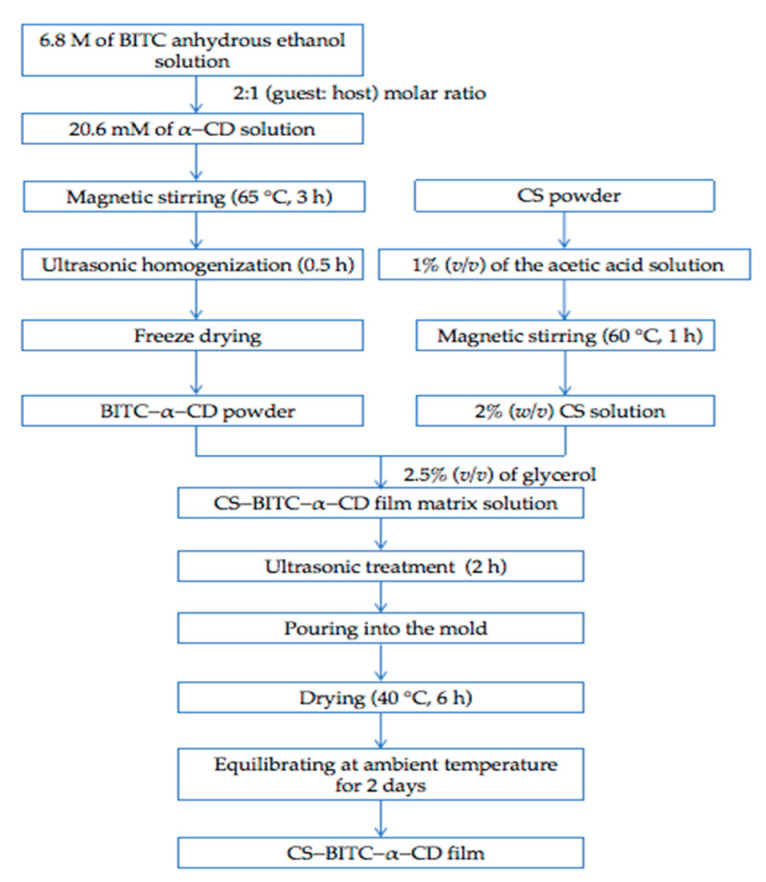
Procedures of fabricating the CS−BITC−α−CD composite film.

**Figure 2 foods-11-02687-f002:**
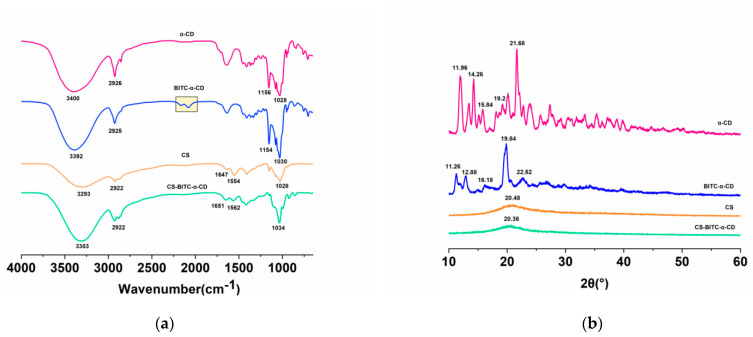
Characterization of inclusion complex and CS−based composite film. (**a**) FTIR spectra of α-CD, BITC−α−CD, CS, and CS−BITC−α−CD; (**b**) XRD patterns of α−CD, BITC−α−CD, CS, and CS−BITC−α−CD.

**Figure 3 foods-11-02687-f003:**
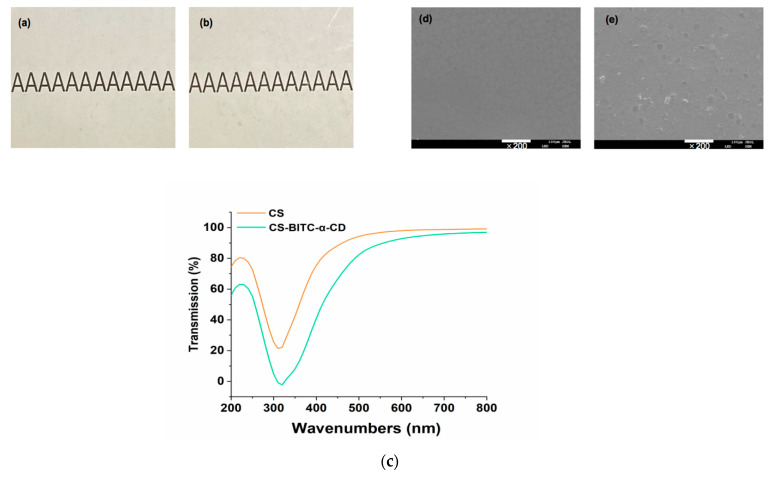
Appearance, light transmittance and surface morphology of the pure CS and CS−based composite films. Appearance of (**a**) pure CS and (**b**) CS−BITC−α−CD composite films; (**c**) UV–vis spectra; surface morphology of (**d**) pure CS and (**e**) CS−BITC−α−CD composite films.

**Figure 4 foods-11-02687-f004:**
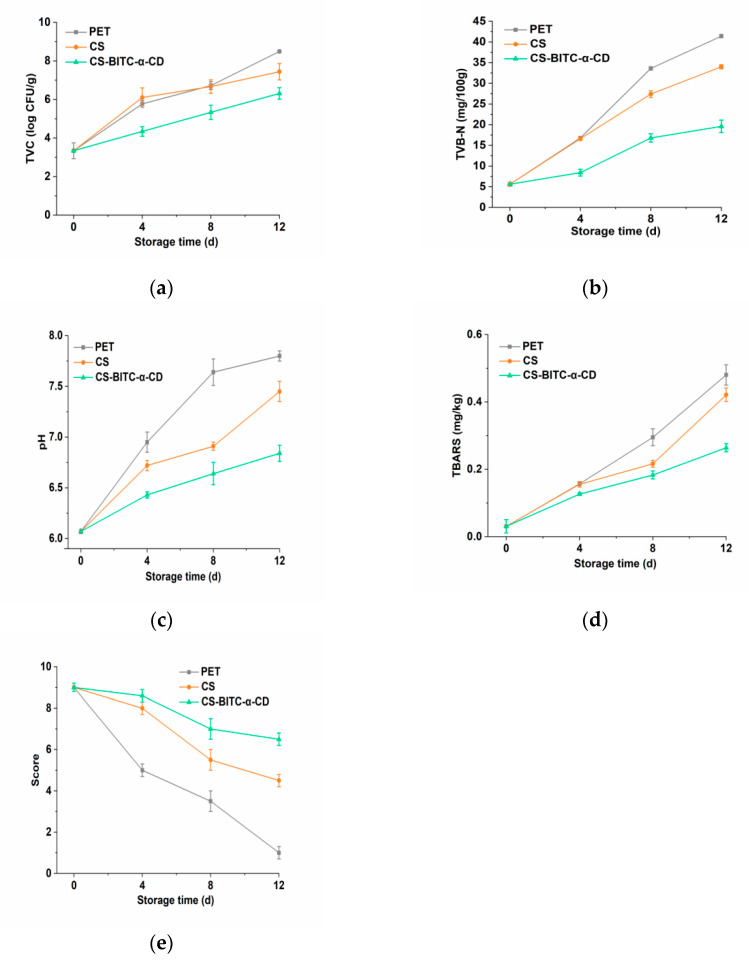
Changes of beef samples wrapped with pure CS and CS−based composite films during storage. (**a**) TVC; (**b**) TVB−N; (**c**) pH; (**d**) TBARS; (**e**) sensory evaluation.

**Table 1 foods-11-02687-t001:** Moisture content (MC), water solubility (WS), and water vapor permeability (WVP) of CS and CS−based composite films.

Samples	MC (%)	WS (%)	WVP ×10^−12^ (g/m·s·Pa)
CS	47.7 ± 6.3 ^a^	41.4 ± 6.3 ^a^	4.2 ± 0.1 ^a^
CS−BITC−α−CD	44.7 ± 1.5 ^a^	48.3 ± 6.4 ^a^	3.9 ± 0.1 ^a^

Values with the same letter in a column indicate no statistical significance (*p* > 0.05).

**Table 2 foods-11-02687-t002:** Antibacterial and antioxidant capacities of CS and CS−based composite films.

Samples	*S*. *Typhimurium* Growth Inhibition (%)		DPPH Scavenging (%)
	12 h	24 h	
CS	4.2 ± 0.3 ^b^	1.3 ± 0.1 ^b^	11.4 ± 0.4 ^b^
CS−BITC−α−CD	31.0 ± 0.4 ^a^	53.3 ± 0.1 ^a^	19.8 ± 0.2 ^a^

Values with different letters in a column indicate statistical significance at *p* < 0.05.

## Data Availability

The data will be shared on reasonable request to the corresponding author.

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
