# Peer review of "A Bioactive Chitosan−Based Film Enriched with Benzyl Isothiocyanate/α−Cyclodextrin Inclusion Complex and Its Application for Beef Preservation"

_foods, 2022, doi:10.3390/foods11172687_

Round 1
Reviewer 1 Report
This study investigated a bioactive chitosan-based film enriched with benzyl isothiocyanate/α-cyclodextrin inclusion complex and its application for beef preservation. The object of this work is interesting; However, most of the experiments are characterized by only two treatments, which is not enough, and the incorporation of different concentrations of BITC-α-CD in the CS film is required.
Moreover, some analyses did not show any significant difference, such as film appearance, MC, WC, and WVP, and other results of the research are too predictive, such as antibacterial and antioxidant properties.
Here are some more tips:
Line 137: Add some more details for SEM analysis, like magnification, voltage, etc.
Line 155: S. Typhimurium should be italic; please check other manuscript parts.
Line 217: It is better to add the FTIR spectrum of BITC to this study.
Line 247-249: For better conclusions, it is recommended to calculate the percentage of relative crystallinity.
Line 257-260: You must mention the photographic conditions in the materials and methods section; or/and use the "HunterLab" test to evaluate the a*, b*, and L* parameters.
Line 269-270: Please report the range of transmittance value.
Line 272: "The previous study ..." add the reference.
Line 313: Add some related articles.
Line 341: Table 2; Please use the letter "a" for the largest value.
Reviewer 2 Report
1. L15- please remove the foodstuff word
2. L16 - "remove "composed inclusion complex"
3. L15-16 - please, if possible, rewrite and add storage details of beef and quality analysis and the package details.
4. L25 - all of which were lower to the value...... is author meant lower is best; please rewrite for clear understanding.
5. L39 - Are ITCs safe, GRAS listed?
6. L49-50 - rewrite for better understanding.
7. In the material and method section, I recommend authors to provide illustrations for the procedures of fabricating the composite films.
8. Details given in the paper for analysis are minimal; please provide elaborated details for all methods.
9. Add an additional section heading beef quality analysis, and under the section, list all the beef quality tests.
10. Author mentions the amorphous status; is it provide good barrier property to the films?
11. Morphology results and their explanation is unclear; please provide a clear explanation about what changes the authors are looking for from the results. And those finding, how is impacting the quality?
12. L295-301 - Could the author using FTIR and XRD results to relate this phenomenon?
13. L311-313 - this phrase is not clear. Could you please report those changes in the results are good or bad for the film?
14. Why antioxidant activity of the package is very low? Chitosan know to have good antioxidant and antimicrobial activity, but in this study, what authors have reported as an activity for chitosan is very low; what could be the cause? Could you please explain or compare similar results from the other study?
15. Section 3.5 - please provide discussion and support from previous findings for the authors' results. At present, not enough discussion is provided.
Reviewer 3 Report
The manuscript contains few sample treatments (2-3 formulations). However, it can provide progress to the field. In general, the manuscript is well written. However, there are some points to reconsider.
Introduction is well written.
Fig. 2 Scale bars are too small to see the number.
L306-307 -> But the stat is insignificant. The discussion should be non-significant different (which means that these values were not different.
L 313 Add more discussion e.g., The WVP values of the films depended on several factors including microstructures, density of the matrices, hydrophilicity, crystallinity, and interaction between polymer components (doi.org/10.1016/j.fpsl.2021.100787; doi.org/10.3390/polym13234192).
Table 1 Recheck unit of WVP (with Eq. 3).
L368 Add more discussion e.g., Similarly, Katekhong (2022) also found inhibition of microorganisms in meat packaged in films containing nitrite which extended shelf-life of meat by more than 12 days (doi.org/10.1016/j.foodchem.2021.131709).
L392 TBARS is lipid oxidation?
Round 2
Reviewer 1 Report
The authors addressed all my suggested comments; this manuscript is now acceptable.
Reviewer 3 Report
The manuscript has been imrpoved.